# Baseline Levels of Vitamin D in a Healthy Population from a Region with High Solar Irradiation

**DOI:** 10.3390/nu13051647

**Published:** 2021-05-13

**Authors:** Alicia García-Dorta, Lillian Medina-Vega, Jacobo Javier Villacampa-Jiménez, Marta Hernández-Díaz, Sagrario Bustabad-Reyes, Enrique González-Dávila, Federico Díaz-González

**Affiliations:** 1Servicio de Reumatología, Hospital Universitario de Canarias, 38320 La Laguna, Spain; alicia.garcia.dorta@gmail.com (A.G.-D.); martahediaz@gmail.com (M.H.-D.); sagrario.bustabad@gmail.com (S.B.-R.); 2Servicio de Análisis Clínicos, Hospital Universitario de Canarias, 38320 La Laguna, Spain; lilianmedinatfe@hotmail.com (L.M.-V.); jacobvilla@msn.com (J.J.V.-J.); 3Departamento de Estadística e Investigación Operativa, Universidad de La Laguna, 38320 La Laguna, Spain; egonzale@ull.edu.es; 4Departamento de Medicina Interna, Dermatología y Psiquiatría, Universidad de La Laguna, 38320 La Laguna, Spain

**Keywords:** vitamin D levels, population-based distribution, healthy adults, high solar irradiation

## Abstract

The use of vitamin D (VitD) supplements has become widespread in the last decade due not only to the dissociation between the blood levels recommended as “optimal” and those shown by the healthy population but also to its presumed beneficial effects on multiple disorders. This work evaluated the levels of 25-hydroxyvitamin D (25(OH)D) in a healthy population of European origin living in a region with high solar irradiation. In serum samples from a population-based study conducted in the Canary Islands, levels of 25(OH)D were analyzed. In 876 individuals who had no history of kidney or malabsorption disorders and, who had not been treated with calcium and/or VitD supplementation, the median 25(OH)D level was 26.3 (5th; 95th percentile, 14.3; 45.8) ng/mL. Notably, 65.4% of the population had 25(OH)D blood levels below 30 ng/mL, 23.4% below 20 ng/mL and 6.4% below 15 ng/mL. Based on the lack of evidence supporting causality between 25(OH)D levels below what is recommended as optimal (≥20 ng/mL, or even ≥30 ng/mL) and major skeletal and non-skeletal diseases, and in light of the distribution of the concentration of this vitamin in healthy adults living under optimal conditions of solar irradiation, it seems reasonable to consider 25(OH)D levels below 20 ng/mL and close to 15 ng/mL as adequate for the general population.

## 1. Introduction

Vitamin D (VitD) is a lipid-soluble hormone that plays a key role in phosphorus and calcium metabolism and bone homeostasis [1]. In humans, it is obtained endogenously by the action of ultraviolet radiation (UVR) and exogenously through dietary intake [2,3]. Exogenous sources of VitD3 include certain foods (notably oily fish, such as trout, salmon, and mackerel), by food fortification in the form of dairy products, orange juice or bread [4] or by the medical prescription of supplements. Endogenous synthesis in the skin is usually efficient and represents an easy and reliable way for most people with sufficient sun exposure to attain adequate VitD levels [3]. It has been described that a full day of sun exposure produces 10,000 to 25,000 IU of VitD [5]. Although it depends on age, skin type and color, the area of exposed skin, season and time of day (reviewed in [6,7]), if sun exposure is adequate, then 10 min of summer sun, without sunscreen, any dietary VitD deficiency is generally less relevant [8]. The form of VitD that circulates in the blood at the highest concentrations and best reflects the overall results of endogenous synthesis and dietary intake is 25(OH)D [9].

It is well established that bone disorders such as rickets and osteomalacia are associated with VitD deficiency [1]. Nonetheless, there is controversy surrounding the optimum levels of this vitamin needed to maintain musculoskeletal health in the adult population [10,11,12,13,14]. Over the last decade, using 25(OH)D supplements has become widespread in Western countries, mainly in response to the observation that basal 25(OH)D levels in many people in the general population [15,16] fail to reach those proposed as optimal by international guidelines, which recommend a threshold value of 20 ng/mL [17]. In contrast, others even aim for ≥30 ng/mL [9]. Another factor encouraging using VitD supplementation stems from observational studies that directly correlate VitD levels with a lower risk of developing a wide range of major non-skeletal human conditions, from depression [18] to cancer [19,20], as well as infections [21,22] and cardiovascular events [23]. However, in recent years, randomized controlled studies have progressively downplayed the potential benefits of VitD concerning these [24,25,26,27,28,29,30] and other non-skeletal health problems [31,32]. In those studies designed to analyze causality, when the effect of VitD supplementation (versus placebo) was analyzed according to baseline VitD levels, no significant benefits were found in individuals with baseline 25(OH)D levels below 20 ng/mL [13,14,24,25,27,29,32]. All this evidence strongly suggests that the current VitD levels recommended as optimal for the general population are overestimated.

The primary objective of this study was to assess the distribution of basal 25(OH)D levels in a healthy population of European origin from an area of year-round sunny warm weather, such as that living in the Canary Islands (28° N latitude). The analyses were overall, by seasonality, and stratified by demographic factors (age, gender, and place of residence). Since obesity has been negatively associated with VitD levels [33,34], the body mass index (BMI) was also analyzed. Results obtained under these optimal conditions could help define normal VitD levels according to a healthy population-based distribution.

## 2. Materials and Methods

### 2.1. Sample Size and Representativeness

For calculating the sample size, we initially analyzed 100 random samples from 1240 patients included in EPIRCAN, a multistage population-based study conducted between May 2004 and September 2005 to assess the prevalence of rheumatic diseases among over-20 year-olds living in the Canary Islands (Atlantic islands 28° N, 16° W). Based on the 2005 census of 1,551,000 people over 20 years of age in the Canary Islands [35], and assuming a standard deviation of 11 ng/mL in line with that in the sample analyzed, a confidence interval of 95% and a precision in the measurement 25(OH)D of 0.7 ng/mL, the required sample size was estimated to be 949 individuals. This cohort was selected by stratified sampling as a function of gender, age group (20–44, 45–64 and ≥65 years old), and place of residence (rural/urban) in line with the population distribution there in 2005 according to the Canary Islands Statistics Institute (ISTAC) [35]. Appendix A sets out the number and percentage of individuals in the sample selected for each stratification variable (age, gender, and place of residence). Information on medical history, including former and current medication, was obtained for everyone in this study. Height and weight were measured, and BMI calculated.

### 2.2. Study Design

This was a cross-sectional population-based observational study to assess 25(OH)D levels in a healthy population of the Canary Islands. Levels of 25(OH)D, calcium and phosphate were measured in 949 blood samples from the EPIRCAN study. A total of 73 individuals (7.7%) with a prior history of kidney failure, dialysis, inflammatory bowel disease, malabsorption, osteoporosis, or with a glomerular filtration rate <60 mL/min/1.73 m^2^ were excluded. Patients on bisphosphonate or calcitonin treatment or calcium and/or VitD supplementation were also excluded. The final sample analyzed included 876 healthy individuals (Table 1). The research was carried out in compliance with the Helsinki Declaration. The study protocol was approved by the Institutional Review Committee at Hospital Universitario de Canarias (Approval number PIET_13 2004). All subjects included in this study provided informed written consent.

### 2.3. Measurement of Blood 25(OH)D Levels

The blood samples from the participants of the EPIRCAN study were stored at −80 °C until use. Levels of 25(OH)D were measured with a chemiluminescence immunoassay (CLIA) run on an ARCHITECT i1000SR analyzer (Abbott, Chicago, IL, USA), while calcium and phosphate levels were measured by absorbance using the Cobas 702c module (Roche Diagnostics, Basel, Switzerland). To rule out significant degradation of the serum samples during storage, we measured the levels of 25(OH)D in fresh blood from a group of 100 individuals living in the northern part of the island of Tenerife between 1 August and 31 October 2019, with a sex ratio of 1:1 and an average age of 49.2 ± 19.6 years, with no medical history of bone, renal or malabsorptive diseases or 25(OH)D supplementation. The median and 5th; 95th percentiles (P_5_; P_95_) of 25(OH)D level in this sample was 27.7 (12.7; 42.9) ng/mL, which was within the range observed in the study population (see below).

### 2.4. Statistical Analysis

Data were stratified by demographic characteristics, comorbidities, and medication use. Continuous variables were described using means and standard deviation or median and percentiles P_5_; P_95_ when not normally distributed, and categorical variables using the frequency and percentage. Groups were compared using Mann–Whitney U and Kruskal–Wallis tests, or Student’s *t*-tests and analysis of variance (ANOVA) for continuous variables and chi-squared tests for categorical variables. A general linear model (ANOVA type II) was used to assess the influence of demographic characteristics on 25(OH)D levels. For 25(OH)D, phosphate and calcium levels, the Kolmogorov–Smirnov test confirmed that the data were not normally distributed and were positively skewed (tail to the right); these problems were resolved by applying a log transformation (*p*-values > 0.2 for all sex * age groups). This allowed us to estimate thresholds for VitD that left (1−α)% of individuals with lower values, by sex * age groups, using the equation $\mathrm{Percentile}\left(1-a\right){\%}_{\;}=\exp\left(\mu +z_{\alpha }\sigma \right)$, where μ and σ are the mean and standard deviation, respectively, of the log-transformed data (Appendix A), zα is the critical point on the standard normal distribution, for which the right–hand tail has an area equal to α exp is the exponential function. Since ln VitD and BMI showed an inverse correlation in patients between 20 and 44 years old (males r = −0.180 *p* = 0.005 slope = −0.01615; females r = −0.218, *p* = 0.001, slope = −0.01698), μ values of VitD for a given BMI different to the mean can be calculated using the formula in Appendix A. In the rest of the age groups * sex, there was no significant relationship between ln VitD and BMI (all *p*-value > 0.253). Similarly, for a given 25(OH)D level, x, it was possible to obtain the corresponding percentile from the expression ϕ((Ln x−μ)/σ), where ϕ is the distribution function of a standardized normal function and *Ln* the natural logarithm. The concordance between percentiles calculated in this way and that of the sample was assessed using Spearman’s correlation and the Student’s *t*-test of one sample (Appendix A). The statistical analysis was carried out using IBM SPSS Statistics for Windows, Version 25.0 (IBM, Armonk, NY, USA) and Microsoft Excel 365.

## 3. Results

### 3.1. Study Population

Table 1 summarizes the demographic characteristics of the individuals excluded (*n* = 73, 7.7%) and included in the analysis. Of the 876 (92.3%) individuals included, 461 (53%) were men and 415 (47%) women, with an overall mean age of 43.3 ± 15.8 years old. The BMI of included individuals was 26.7 ± 4.6 kg/m^2^ for the entire population without significant differences between women (26.5 ± 5.2) and men (26.9 ± 4.1), *p* = 0.23. Classifying by residence, 134 (15%) lived in rural and 742 (85%) in urban areas. In terms of age, 527 (60%) were between 20 and 44 years old, 233 (27%) between 45 and 64 years old, and 116 (13%) were 65 years old or more.

### 3.2. Levels and Distribution of 25(OH)D, Calcium and Phosphate by Gender, Age, Place of Residence and BMI in the Healthy Population

The median (P5; P95) level of 25(OH)D in the healthy population was 26.3 (14.3; 45.8) ng/mL, while calcium and phosphate levels were 9.3 (8.1; 10.2) ng/dL and 3.5 (2.5; 5.2) ng/dL, respectively (Table 1). Analyzing 25(OH)D levels by gender, age and place of residence, we found significant differences between genders (Table 2, Figure 1A) and age groups (Table 2, Figure 1B), but not by place of residence (*p* = 0.218). Levels of 25(OH)D correlated inversely with BMI in the entire population (r = −0.192, *p* < 0.001), mainly due to the 20–44 years old group (r = −1.90, *p* < 0.001).

Table 3 shows that 25(OH)D levels were 2.7 units lower in women than in men (95% CI −4.1; −1.4, *p* < 0.001) and 4.4 units higher in 20 to 44 year-olds (95% CI, 2.3; 6.5, *p* < 0.001) and 2.5 units higher in 45 to 64 year-olds (95% CI 0.4; 4.6, *p* < 0.019) than in the oldest age group (≥65 years). VitD levels decreased by 0.298 ng/mL (95% CI, −0.45; −0.15, *p* < 0.001) for each unit of increase in BMI.

VitD levels obtained after applying a log transformation were normally distributed by sex * age groups, and the percentiles calculated by this transformation and those of the sample only presented slight deviations (Appendix A). This allowed us to determine the percentage of the population with 25(OH)D levels below certain thresholds; Table 4 shows these threshold percentages ranging from 12 to 30 ng/mL, both overall and stratified by age or sex. The 20–44 age group shows the percentiles considering the mean BMI for women (24.92) and men (25.89). For other BMIs, the values can be calculated using the formula in Table 3. Notably, 65.5% of the population had 25(OH)D levels under 30 ng/mL, while 23.4%, 6.4% and 2% of the population had levels below 20, 15 and 12.5 ng/mL, respectively.

Figure 1C shows the distribution of 25(OH)D levels in the overall sample analyzed.

### 3.3. Seasonal Variation in Serum 25(OH)D

During the study, the mean total solar irradiation increased from 4.21 ± 0.67 kWh/m^2^/day during the October to March period to 6.36 ± 0.43 kWh/m^2^/day between April to September [36] and positively correlated with serum a 25(OH)D increment of 2.01 ng/mL (95% IC: 0.73–3.30, *p* = 0.002). Average concentrations of 25(OH)D showed large variations over the different months of the year (Table 5 and Figure 2) with a profile that replicated the shape of the curve of total solar irradiation with a delay of 2 months (Figure 2). The mean serum concentration of 25(OH)D was 44% higher in September (30.51 ± 10.25 ng/mL) than in March (21.78 ± 8.21 ng/mL; *p* < 0.0001). During the third quarter of the year (July to September), the median serum concentration of 25(OH)D, 28.1 (P5-P95:15.5; 52.2) ng/mL was 28% higher than during the first quarter (January–March), 22.0 (P5-P95:10.0; 36.9) ng/mL, *p* < 0.001). During the first quarter of the year (January–March), 12.3% (23/187) of patients had serum concentrations of 25(OH)D lower than 15 ng/mL, 38.0% (70/187) lower than 20 ng/mL, and 78.6% (145/187) had levels lower than 30 ng/mL, compared with the third quarter of the year (July–September), when 4.6% (8/175) had levels below 15 ng/mL, 9.7% (17/175) below 20 ng/mL, and 54.9% (94/175) leveled below 30 ng/mL, *p* < 0.001 (Table 5).

## 4. Discussion

The most relevant findings in this study can be summarized as follows: (1) a healthy population of European origin living in an area with high levels of solar radiation and not taking VitD supplements had a median 25(OH)D level of 26 ng/mL; and (2) based on the population distribution, 65.5% of the healthy individuals had 25(OH)D levels below 30 ng/mL, 23.4% below 20 ng/mL and 5% levels below 14.3 ng/mL. Given that these VitD population distribution data were collected under optimal conditions, together with the absence of studies demonstrating any causal relationship between vitamin D levels (<20 vs. ≥20 ng/mL) and certain skeletal and extraskeletal pathological conditions, it seems reasonable to consider 25(OH)D levels below 20 ng/mL and very close to 15 ng/mL as adequate for healthy adult populations. We believe that a downward readjustment of the recommended “optimal” VitD levels for healthy populations to these proposed new limits will have important implications not only for healthcare costs [37], and safety [38,39,40], avoiding over diagnosed and overtreated for a VitD “deficiency”, but also for the design of trials examining the benefits of VitD supplementation in human diseases.

Studies in the United States of America, France and China have found 25(OH)D levels <30 ng/mL in 75% [15], 80% [16] and 83% [41] of the population, respectively, figures that are overall higher than those in our study (65.5%). This could be explained by differences between ethnic groups and/or levels of local solar irradiation. Genetic studies of the current population of the Canary Islands have shown that 90% of the paternal lineage is of European origin [42]. Sunlight exposure is key to synthesize VitD [3]. Exposure of the face and arms to 3–7 min of sunlight each day in areas with solar radiation of 1.8 kWh/m^2^/day in the summer months allows synthesizing 400 to 1000 IUs of VitD, equivalent to the recommended daily dose of this vitamin [43]. In this regard, several studies on the effect of solar irradiation on 25(OH)D production in Europeans have been performed in the Canary Islands [44,45]. Between 1983 and 2005, the Canary Islands had a mean daily solar radiation level of 5.23 kWh/m^2^/day, much higher than that of European cities, such as London (2.88 kWh/m^2^/day), Paris (3.25 kWh/m^2^/day) or Rome (4.5 kWh/m^2^/day) [36]. In our population-based study, VitD levels varied seasonally following changes in solar irradiation, with a time lag of approximately 2 months, in a range like previously observed [46]. The percentage of individuals with VitD levels below 15 ng/mL remained stable at around 5% throughout the year, except during the first quarter (January–March), when this percentage reached 12% of the population.

While there is consensus on the levels of 25(OH)D that define clinical deficiency (<4 ng/mL), there is no agreement on the optimal levels of this prehormone in the general population [47]. In 2003, the World Health Organization defined VitD deficiency as 25(OH)D levels below 10 ng/mL and insufficiency as levels below 20 ng/mL [48]. Since then, however, other reference ranges have been established [9,17], and in the last 10 years, most clinical laboratories have recommended that the general population aim for levels above 30 ng/mL. The rationale for establishing this limit was based on reports that 25(OH)D levels below 30 ng/mL were associated with increases in parathyroid hormone [49,50,51]. However, there is evidence suggesting that the relationship between parathyroid hormone and 25(OH)D is inconsistent, and there is no absolute threshold level of serum 25(OH)D at which PTH levels rise [50,51], and hence, should not be used as the basis for determining optimal VitD levels in the general population [17].

Regarding the role of VitD in skeletal health, it has been suggested that 25(OH)D levels >12.5 ng/mL are sufficient for maintaining good musculoskeletal health in middle-aged women [52]. In this context, an ancillary study of the VITAL trial, which randomized almost 26,000 men and women older than 50 years to 2000 IU of VitD or placebo, included a subcohort of 687 participants older than 50 years whose bone mineral density (BMD) at the spine, hip, and whole-body were assessed at baseline and over a 2 year period. Results showed no effect on BMD after 2 years of VitD supplementation either in the whole population or in subgroups defined by baseline 25(OH)D levels <15 ng/mL vs. >15 ng/mL or even at <10 ng/mL vs. >10 ng/mL [14]. Similar negative results regarding the benefit of VitD supplementation (in the entire population or in the same subgroups of baseline 25(OH)D) in the risk of falls were provided by the VITAL trial [53]. The role of VitD supplementation on non-vertebral fractures and falls was analyzed both as secondary and post hoc outcomes in the VIDA trial, a randomized placebo-controlled study that included more than 5000 men and women older than 50 years [54]. After a mean of 3.4 years of follow-up, the study concluded that VitD supplementation did not prevent falls or fractures in healthy adult populations. Interestingly, when the analysis was performed in subgroups defined by 25(OH)D at baseline, VitD supplementation in individuals with less than 10 ng/mL of 25(OH)D levels did not result in reduced fracture risk [13].

Defining the threshold of normality for VitD has key implications not only for healthcare costs [37]—i.e., avoiding unnecessary VitD screening and supplementation [55]—but also for the proper design of clinical trials to evaluate the impact of VitD supplementation on different humans diseases. A recent randomized clinical trial on type 2 diabetes [56] prevention illustrated this latter topic very well. The results showed that VitD supplementation does not affect preventing type 2 diabetes. Baseline 25(OH)D levels of the patients included clearly exceeded the 20 ng/mL threshold: 28.2 ng/mL. A relevant conclusion of this work is that VitD supplementation in already repleted patients does not affect type 2 diabetes prevention. A post hoc analysis of this study showed that only those individuals with a baseline 25(OH)D level <12 ng/mL benefited from VitD supplementation [56]. Studying the effects of VitD only in deficient subjects greatly increases the likelihood of showing a benefit from supplementation since the effect of a threshold nutrient is not likely to emerge if both the control and supplementation groups have sufficient levels of 25(OH)D at baseline [47].

Regarding gender differences, in our study, 25(OH)D levels were significantly higher in men than in women. Similar patterns have been found in Greek schoolchildren [57] and in a Chinese adult population [41]. Concerning age, our results indicate a significant trend towards decreases in 25(OH)D levels with age, in line with previous reports [58,59]. The inverse association between 25(OH)D levels and BMI found in this study is supported by previous studies [33,34].

Our study has certain strengths and weaknesses regarding its objective to assess the population distribution of basal 25(OH)D levels: (1) this is a healthy population-based study based on a healthy population with sufficient power for determining the levels of this vitamin to within 0.7 ng/mL; (2) measurements were made on a representative sample of residents in the Canary Islands, people of European origin, who enjoy high levels of solar irradiation; (3) in our study we have proactively ruled out interference from VitD supplementation, the most potent source of 25(OH)D, and samples were obtained 15 years ago were VitD food fortification was less common; (4) 25(OH)D levels were determined using an automated CLIA, a cost-effective and time-efficient method [60], but less accurate and reproducible than liquid chromatography-tandem mass spectrometry (LC–MS/MS), the best method to quantify VitD [61]; and (5) although clothing in the Canary Islands commonly leaves the arms uncovered and that the most frequent skin phototypes are II-IV, being the III the most common, these and other parameters with a smaller impact on 25(OH)D levels than VitD supplementation, such as using sunscreens or time outdoors, were not considered in this work.

## 5. Conclusions

Given the evidence supporting that VitD supplementation in adult individuals with baseline 25(OH)D levels <15 ng/mL does not result in improved musculoskeletal health [13,14,53], and the fact that a causal relationship of VitD with numerous non-skeletal diseases has not yet been demonstrated [24,25,31,62], and in light of the distribution of the concentration of this vitamin in healthy adults living under optimal conditions of solar irradiation, it seems reasonable to consider 25(OH)D levels below 20 ng/mL and close to 15 ng/mL as optimal, at least for the general population of European origin.

## Figures and Tables

**Figure 1 nutrients-13-01647-f001:**
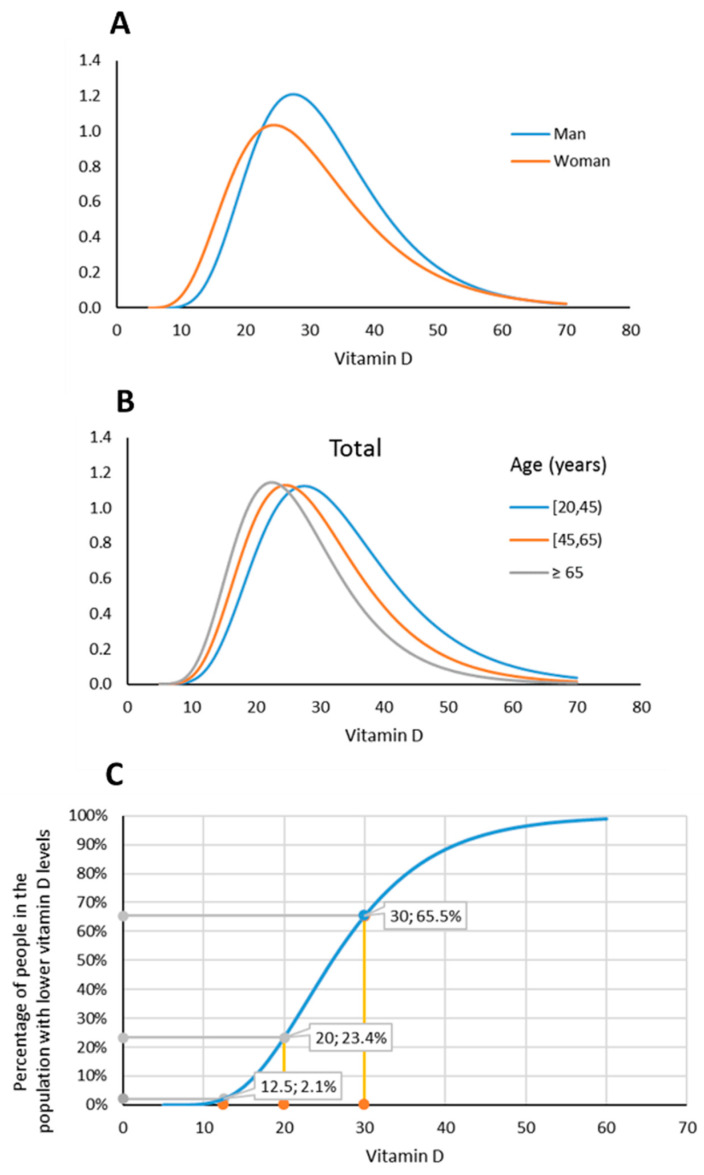
(**A**) Density function of VitD concentration (ng/mL) by gender. (**B**) Density function of VitD concentration (ng/mL) by age groups. (**C**) Distribution of percentiles of VitD concentration.

**Figure 2 nutrients-13-01647-f002:**
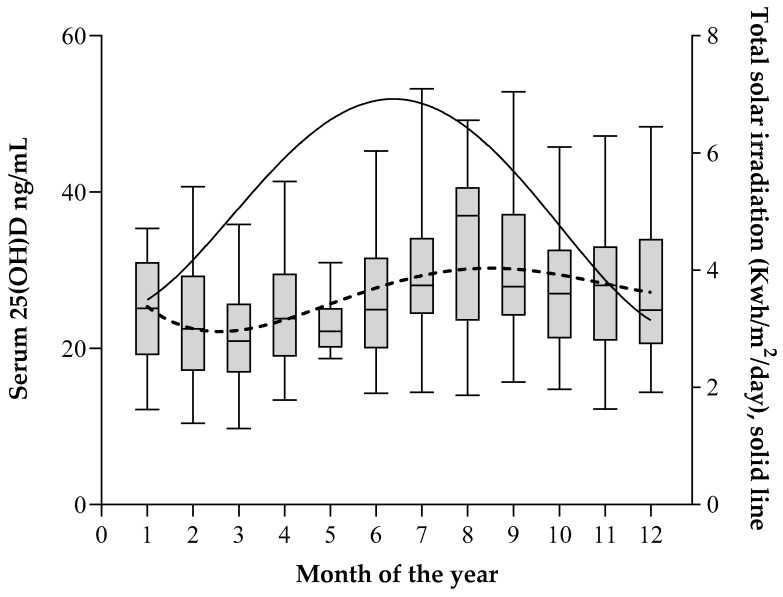
Box plot graph showing the serum 25(OH)D levels during the different months of the year. The values represent medians (horizontal line), interquartile ranges (box) and P5 and P95 percentile (whiskers). The solid line represents the variation of mean total solar irradiation in the Canary Islands during the time frame of the study.

**Table 1 nutrients-13-01647-t001:** Demographic characteristics and levels of VitD, calcium and phosphate in the included (healthy) and excluded individuals.

	Healthy Individuals *n* = 876 (92.3%)	Excluded * *n* = 73 (7.7%)	*p*	Total *n* = 949
Sex			<0.001	
Male	461 (53%)	13 (18%)		474 (50%)
Female	415 (47%)	60 (82%)		475 (50%)
Age (years)	43.3 ± 15.8	59.1 ± 14.9	<0.001	44.5 ± 16.3
Age (years)			<0.001	
20–44	527 (60%)	10 (14%)		537 (57%)
45–64	233 (27%)	33 (45%)		266 (28%)
≥65	116 (13%)	30 (41%)		146 (15%)
Area			0.478	
Rural	134 (15%)	14 (19%)		148 (16%)
Urban	742 (85%)	59 (81%)		801 (84%)
BMI (kg/m^2^)	26.74 ± 4.67	27.79 ± 4.76	0.070	26.82 ± 4.69
25(OH)D (ng/mL)	26.30 (14.30; 45.84)	23.70 (11.50; 47.7)	0.351	26.20 (14.20; 45.8)
Phosphate (mg/dL)	3.50 (2.54; 5.25)	3.48 (2.35; 4.65)	0.425	3.50 (2.53; 5.19)
Calcium (mg/dL)	9.34 (8.15; 10.28)	9.50 (8.13; 11.12)	0.072	9.35 (8.15; 10.31)

Data are expressed as *n* (%), mean ± SD or median (P_5_; P_95_). * Individuals with a prior history of kidney failure, inflammatory bowel disease, malabsorption, or osteoporosis, or, who had been treated with calcium and or VitD supplements, bisphosphonates or calcitonin, or on dialysis were excluded. BMI: body mass index.

**Table 2 nutrients-13-01647-t002:** Distribution of the sample of healthy patients by gender and age.

	Male *n* = 461	Female *n* = 415	*p*		Age (Years)		*p*	Total *n* = 876
				20–44 *n* = 527	45–64 *n* = 233	≥65 *n* = 116		
Sex							0.558	
Male				271 (51%)	130 (56%)	60 (52%)		461 (53%)
Female				256 (49%)	103 (44%)	56 (48%)		415 (47%)
Age (years)	43.9 ± 15.9	42.7 ± 15.7	0.259					43.3 ± 15.8
Age (years)			0.558					
20–44	271 (59%)	256 (62%)						527 (60%)
45–64	130 (28%)	103 (25%)						233 (27%)
≥65	60 (13%)	56 (13%)						146 (13%)
Area			0.854				0.459	
Rural	72 (16%)	62 (15%)		74 (14%)	39 (16%)	21 (18%)		134 (15%)
Urban	389 (84%)	353 (85%)		453 (86%)	194 (84%)	95 (82%)		742 (85%)
BMI (kg/m^2^)	26.92 ± 4.13	26.53 ± 5.21	0.235	25.42 ± 4.39	28.68 ± 4.62	28.14 ± 4.11	<0.001	26.74 ± 4.67
25(OH)D (ng/mL)	27.4 (15.8; 46.7)	25.0 (12.8; 45.3)	<0.001	27.4 (15.6; 48.1)	25.0 (13.4; 43.7)	23.0 (12.4; 36.6)	<0.001	26.3 (14.3; 45.8)
Phosphate (mg/dL)	3.48 (2.4; 4.8)	3.51 (2.6; 6)	0.180	3.58 (2.6; 5.7)	3.46 (2.5; 5.)	3.23 (2.5; 4.8)	0.001	3.5 (2.5; 5.2)
Calcium (mg/dL)	9.35 (8.; 10.3)	9.32 (8.2; 10.2)	0.584	9.36 (8.2; 10.2)	9.30 (8.1; 10.4)	9.19 (7.9; 10.3)	0.031	9.3 (8.1; 10.2)

Data expressed as mean ± SD, *n* (%) or median (P_5_; P_95_).

**Table 3 nutrients-13-01647-t003:** General linear model for VitD level concerning the demographic factors considered.

Parameter	Coefficient	SE	Student’s *t*	*p*	95% CI
Intersection	33.607	2.326	14.447	<0.001	(29.041; 38.174)
Age (ref ≥ 65)					
20–44	4.372	1.045	4.183	<0.001	(2.320; 6.424)
45–64	2.483	1.056	2.350	0.019	(0.409; 4.557)
Female sex	−2.689	0.719	−3.742	<0.001	(−4.099; −1.278)
Rural area	−1.438	1.166	−1.234	0.218	(−3.727; 0.850)
BMI (kg/m^2^)	−0.298	0.075	−3.969	<0.001	(−0.446; −0.151)

BMI: body mass index; SE: standard error; 95% CI: 95% confidence interval.

**Table 4 nutrients-13-01647-t004:** Percentage of healthy individuals (percentile) with 25(OH)D levels below the values in the first column.

Baseline 25(OH)D (ng/mL)	Male (Age, Years)			Female (Age, Years)			Gender		Age (Years)			Total
	20–44 *	45–64	≥65	20–44 *	45–64	≥65	Male	Female	20–44 *	45–64	≥65	
12	0.3%	1.3%	0.8%	2.1%	3.2%	7.4%	0.6%	3.2%	1.0%	2.1%	3.6%	1.6%
13	0.6%	2.3%	1.6%	3.4%	5.2%	11.2%	1.1%	5.0%	1.7%	3.5%	5.8%	2.7%
14	1.2%	3.7%	2.9%	5.2%	7.9%	15.8%	2.0%	7.3%	2.9%	5.5%	8.8%	4.3%
15	2.0%	5.6%	4.7%	7.4%	11.2%	21.1%	3.3%	10.2%	4.4%	8.0%	12.4%	6.4%
16	3.2%	8.1%	7.2%	10.0%	15.0%	26.8%	5.0%	13.5%	6.4%	11.1%	16.5%	8.9%
17	4.8%	11.1%	10.3%	13.1%	19.4%	32.9%	7.2%	17.2%	8.8%	14.7%	21.2%	12.0%
18	6.9%	14.6%	14.0%	16.5%	24.1%	39.0%	9.9%	21.3%	11.6%	18.8%	26.3%	15.4%
19	9.5%	18.4%	18.2%	20.2%	29.1%	45.1%	13.1%	25.6%	14.9%	23.2%	31.6%	19.3%
20	12.5%	22.7%	22.9%	24.2%	34.3%	51.0%	16.7%	30.1%	18.5%	27.8%	37.0%	23.4%
21	16.0%	27.2%	28.0%	28.3%	39.5%	56.5%	20.7%	34.6%	22.4%	32.7%	42.4%	27.7%
22	19.8%	31.8%	33.2%	32.5%	44.6%	61.7%	25.0%	39.2%	26.5%	37.5%	47.7%	32.2%
23	23.9%	36.6%	38.6%	36.8%	49.6%	66.5%	29.5%	43.7%	30.7%	42.4%	52.8%	36.7%
24	28.2%	41.3%	43.9%	41.0%	54.4%	70.8%	34.1%	48.1%	35.1%	47.2%	57.6%	41.3%
25	32.7%	46.0%	49.2%	45.2%	58.9%	74.7%	38.7%	52.3%	39.4%	51.8%	62.2%	45.7%
26	37.3%	50.5%	54.2%	49.2%	63.1%	78.2%	43.4%	56.4%	43.7%	56.2%	66.4%	50.0%
27	41.9%	54.9%	59.0%	53.2%	67.1%	81.3%	47.9%	60.2%	47.9%	60.4%	70.2%	54.2%
28	46.4%	59.0%	63.5%	56.9%	70.7%	83.9%	52.3%	63.8%	52.0%	64.3%	73.7%	58.2%
29	50.8%	62.9%	67.7%	60.4%	74.0%	86.3%	56.6%	67.1%	55.9%	67.9%	76.9%	61.9%
30	55.0%	66.6%	71.5%	63.8%	77.0%	88.3%	60.6%	70.3%	59.7%	71.3%	79.8%	65.5%

* Values for mean BMI of 24.92 for females and 25.89 for males.

**Table 5 nutrients-13-01647-t005:** Serum levels of 25(OH)D during the different quarters of the year.

Quarters Months	Number of Subjects	25(OH)D ng/mL	25(OH)D ng/mL	25(OH)D ≤ 15 ng/mL	25(OH)D ≤ 20 ng/mL	25(OH)D ≤ 30 ng/mL
Q1 January–March	187	23.1 ± 8.4	22.0 (10.0; 37.0)	23 (12.3%)	71 (38.0%)	147 (78.6%)
Q2 April–June	275	27.0 ± 9.1	25.4 (14.0; 44.0)	14 (5.1%)	67 (24.4%)	183 (66.5%)
Q3 July–September	175	31.1 ± 10.4	28.1 (15.3; 52.2)	8 (4.6%)	17 (9.7%)	96 (54.9%)
Q4 October–December	239	28.8 ± 10.6	27.5 (15.3; 47.6)	10 (4.2%)	39 (22.1%)	144 (60.3%)
All	876	27.6 ± 10.0	26.3 (14.3; 45.8)	55 (6.3%)	194 (22.1%)	570 (65.1%)

Data expressed as mean ± SD, median (P_5_; P_95_) or *n* (%).

## Data Availability

The datasets generated and/or analyzed during the current study are available from the corresponding authors upon reasonable request.

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
