# Peer review of "Baseline Levels of Vitamin D in a Healthy Population from a Region with High Solar Irradiation"

_nutrients, 2021, doi:10.3390/nu13051647_

Round 1

Reviewer 1 Report

In this manuscript. Garcia-Dorta et al have assessed the levels of 25(OH)D in a healthy population of the Canary Islands. They have found that the mean 25(OH)D levels of this healthy cohort are significantly lower than what is considered as the lower limit of the normal range. This is a well-conducted study with sound methodology, good presentation of the results and very interesting conclusions. I have no specific comments or suggestions.   

Author Response

We appreciated the positive appraisal and constructive comments of the reviewer. 

Reviewer 2 Report

Overall I found this study quite interesting. I am a systems engineer, not an MD, and my research is on quantitative analysis of healthcare, with a particular focus on economic problems. Hence I found it refreshing to read about a study that took into account the costs of over-treating, especially such a minor issue. Research that casts doubt on consensus views without adequate empirical support are especially welcome in any realm that values transparency and scientific rigor.

The study was observational, which is as it should be, since there are obvious ethical issues with RCTs. Healthy adults having access to plenty of sunlight is the population of interest, admittedly limited to those of European descent.

Studies that focus on small populations of patients with specific ailments may find that particular diseases may be influenced by vitamin D at higher levels, in spite of the  lack of effect in healthy adults. This would not be likely to show up in a study of healthy adults (unless the sample size grew enormous), and perhaps this point may be made somewhat stronger. However, I did not find that the authors claimed too much in their discussion. 

I had a few relatively minor editorial suggestions:

Lines 125-128: "This allowed us to estimate thresholds for VitD that left (?−?)% of 125 individuals with lower values, by sex*age groups, using the equation ??−?=???⁡(?+126 ???), where ? and ? are the mean and standard deviation, respectively, of the log-trans-127 formed data (Table S2), ??⁡is the critical point of a standardized normal distribution with 128 an area to the right of ? and ??? is the exponential function."
This is mostly fine, except that the description of z is misleading. Since alpha itself is the size of an area under the standard normal curve (to the right of z), z is not a point "with an area to the right of " alpha. Instead, it is the critical point on the standard normal distribution, for which the right hand tail has area equal to alpha. Or equivalently, such that Prob(Z>z)=alpha
I'd also suggest that you use something other than P to indicate thresholds for VitD, since any time you see a P in a statistics discussion, the mind leaps to probabilities and proportions, and that is not what was intended.

lines 231-234: "We believe that a downward readjustment of the 231
recommended "optimal" VitD levels for healthy populations to these proposed new limits will have important implications not only for healthcare costs [37], and safety [38-40], avoiding over diagnosed and overtreated for a VitD “deficiency”, but also for the design of trials examining the benefits of VitD supplementation in human diseases."
>>> It is not clear that this will save a great deal in terms of cost [37] or safety [38-40]; quantifying the effects would be welcome, but may be beyond the scope of the article. For instance, [37] has the cost per screening at GBP 9.86; while the cost of a supplement of VitD per year is not stated (implicitly because it is borne by the patient?). 

lines 254-255: "While there is consensus on the levels of 25(OH)D that define clinical deficiency (<4 254 ng/mL), there is no agreement on the optimum levels of this prehormone in the general 255 population [47]."
This is being picky, but "optimum" should be changed to "optimal", unless the authors think that achieving some "optimum" level of Vitamin D in and of itself serves a goal. Note also that the correct term, "optimal", was used in the commented quote immediately above. Continuing along somewhat similar lines, why does there actually have to be any optimal level of the pre-hormone? Many inputs into metabolic pathways merely have to exceed some minimal sufficient level. The interesting question here is to identify that minimal sufficient level better than had been done before.

Author Response

We also appreciated the positive appraisal and constructive comments of the reviewer. 

Following he/her suggestions: 

- Lines 125-128: .... This is mostly fine, except that the description of z is misleading. Since alpha itself is the size of an area under the standard normal curve (to the right of z), z is not a point "with an area to the right of " alpha. Instead, it is the critical point on the standard normal distribution, for which the right hand tail has area equal to alpha. Or equivalently, such that Prob(Z>z)=alpha. 

We appreciate and agree with this comment. In the revised version of the manuscript, lines 128 and 129 now reads: "...?? is the critical point on the standard normal distribution, for which the right hand tail has area equal to ?". 

- I'd also suggest that you use something other than P to indicate thresholds for VitD, since any time you see a P in a statistics discussion, the mind leaps to probabilities and proportions, and that is not what was intended.

To avoid confusion, we have followed this reviewer´s recommendation: P has been changed to Percentile. In the revised manuscript, line 126 now reads: "...., Percentile (1-α)% =...”

-lines 231-234:....It is not clear that this will save a great deal in terms of cost [37] or safety [38-40]; quantifying the effects would be welcome, but may be beyond the scope of the article. For instance, [37] has the cost per screening at GBP 9.86; while the cost of a supplement of VitD per year is not stated (implicitly because it is borne by the patient?)

This point raised by the reviewer is very important, although, as he mentions, it is beyond the scope of this article. We so disagree with the reviewer, however, that a reduction in VitD reference levels cannot lead to appreciable savings in health care costs.  
Approximately 10 million euros are spent annually in Spain in direct costs for 25(OH)D determinations, and between 60-70 million euros on VitD supplements (in Spain the cost of this vitamin is covered by the national health insurance). Changing the optimal 25(OH)D levels from the 30 ng/ml used in most clinical laboratories in Spain by the 15 ng/ml we propose in our work would mean a decrease in 90% of the population requiring supplementation. This is a potential savings of 45-55 million euros. These are not negligible figures, especially in the current situation of severe economic recession that has resulted from the COVID pandemic. 
Undoubtedly this point would require a more careful analysis than this simple approximation. 

lines 254-255:.....This is being picky, but "optimum" should be changed to "optimal", unless the authors think that achieving some "optimum" level of Vitamin D in and of itself serves a goal. Note also that the correct term, "optimal", was used in the commented quote immediately above. Continuing along somewhat similar lines, why does there actually have to be any optimal level of the pre-hormone? Many inputs into metabolic pathways merely have to exceed some minimal sufficient level. The interesting question here is to identify that minimal sufficient level better than had been done before.

Following the reviewer's suggestion, line 256 of the revised manuscript now reads:  "....there is no agreement on the optimal levels of this prehormone ..."